# Protective Effect of Foxtail Millet Protein Hydrolysate on Ethanol and Pyloric Ligation-Induced Gastric Ulcers in Mice

**DOI:** 10.3390/antiox11122459

**Published:** 2022-12-14

**Authors:** Bowei Zhang, Xiaoxiao Rao, Yunhui Zhang, Weijia Dai, Yingchuan Xu, Congying Zhao, Zhenjia Chen, Jin Wang, Dancai Fan, Xiaowen Wang, Shuo Wang

**Affiliations:** 1Tianjin Key Laboratory of Food Science and Health, School of Medicine, Nankai University, Tianjin 300071, China; 2College of Food Science and Engineering, Shanxi Agricultural University, Taiyuan 030801, China

**Keywords:** foxtail millet protein hydrolysates, gastric ulcer, nitric oxide (NO), peptides, gastric mucosal mucus

## Abstract

Foxtail millet has been traditionally considered to possess gastroprotective effects, but studies evaluating its use as a treatment for gastric ulcers are lacking. Here, we assessed the antiulcer effects of foxtail millet protein hydrolysate (FPH) and explored its mechanism by using blocking agents. In a mouse model of ethanol-induced gastric ulcers, pretreatment with FPH reduced the ulcerative lesion index, downregulated the expression of inflammatory cytokines in the gastric tissue, increased the activity of antioxidant enzymes, and improved the oxidative status. FPH increased constitutive the activity of nitric oxide synthase (cNOS), NO levels, and mucin expression in gastric mucosa, and inhibited the activation of the ET-1/PI3K/Akt pathway. In a mouse model of pyloric ligation-induced gastric ulcers, FPH inhibited gastric acid secretion and decreased the activity of gastric protease. Pretreatment of mice with the sulfhydryl blocker NEM and the NO synthesis inhibitor L-NAME abolished the gastroprotective effect of FPH, but not the KATP channel blocker glibenclamide and the PGE2 synthesis blocker indomethacin. Among the peptides identified in FPH, 10 peptides were predicted to have regulatory effects on the gastric mucosa, and the key sequences were GP and PG. The results confirmed the gastroprotective effect of FPH and revealed that its mechanism was through the regulation of gastric mucosal mucus and NO synthesis. This study supports the health effects of a millet-enriched diet and provides a basis for millet protein as a functional food to improve gastric ulcers and its related oxidative stress.

## 1. Introduction

Gastric ulcers are a digestive system disease with high rates of recurrence. The lifetime prevalence of gastric ulcers is approximately 5–10%, and the annual incidence rate is 0.1–0.3% [1]. The pathological manifestations are gastric mucosal injury, erosion, ulcers, and bleeding. Patients often present with epigastric pain, dull pain, nausea, vomiting, acid regurgitation, etc. The long-term development of the disease can lead to acute complications, such as gastrointestinal bleeding and gastric perforation, which could seriously affect the health of patients [2]. The pathogenesis of gastric ulcers is related to smoking, persistent use of non-steroidal anti-inflammatory drugs, *Helicobacter pylori* infection, alcohol abuse, stress, and hypersecretion of gastric acid. Dysregulation of ATP-sensitive K^+^ (KATP) channels and endogenous protective agents such as mucus, nitric oxide, prostaglandins, and growth factors are also important causes [3].

At present, the main clinical treatment for gastric ulcers is drug therapy. The available treatment options for active gastric ulcers include antacids, cytoprotective agents, proton pump inhibitors, H2 histamine receptor antagonists, and a combination of antibacterial drugs [1]. However, these drugs are associated with multiple side effects, including poor ulcer healing and ulcer recurrence, resulting in a huge economic burden on patients and public health systems [4]. Thus, it is of great significance to explore effective and safe gastroprotective agents derived from natural resources. Currently, several edible resources have been found to help improve gastric ulcers [5,6,7].

Foxtail millet (*Setaria italica*) is widely grown in Asia and Africa, and is the most widely consumed coarse grain in China. The protein content of millet is about 11–18% [8], of which the highest content is albumin, followed by gliadin, globulin, and gluten. Millet protein contains 19.89% branched-chain amino acids, 10.2% aromatic amino acids, and 10.2% proline [9]. The component analysis indicates that millet protein may have unique physicochemical properties and physiological functions. Millet is rich in various essential amino acids. Except for the low content of lysine, the other seven amino acids are abundant in millet, especially tryptophan and methionine. The content of each amino acid is higher than the value recommended by FAO/WHO, indicating that millet can be used as a high-quality vegetable protein source. Previous studies have indicated the anti-diabetic [10], antioxidant [11,12], and anti-inflammatory effects of foxtail millet [13]. Our previous study found that a diet rich in millet was protective against colitis-related colorectal cancer [14]. It is worth noting that foxtail millet has been traditionally considered to benefit the stomach. A previous study has suggested the antiulcer activity of foxtail millet [15]. However, the active ingredient of foxtail millet remains to be explored. The hydrolytes of foxtail millet protein have been reported to exhibit anti-hypertensive and anti-diabetic activities [16]. Millet-derived bioactive peptides have been demonstrated to have antioxidant and antifungal activities [17], and could inhibit the secretion of proinflammatory cytokines in RAW264.7 cells in vitro [18]. In addition, we found that millet protein hydrolysate, which is the active ingredient of foxtail millet, has protective effects against intestinal mucosal damage and has inhibitory effects on inflammatory responses [19]. These results suggest that millet protein may have protective effects against gastric ulcers, but its activity has not been studied.

Based on previous research, this study aimed to evaluate the gastroprotective effect of foxtail millet protein hydrolysate (FPH) on ethanol- and ligation-induced gastric ulcers in mice, and to investigate the underlying mechanisms by using blocking agents.

## 2. Materials and Methods

### 2.1. Chemical and Materials

Antibodies for Western blotting were purchased from Cell Signaling Technology (Shanghai, China). Gastrin (Gastrin), malondialdehyde (MDA), glutathione peroxidase (GSH-PX), superoxide dismutase (SOD), catalase (CAT), nitric oxide (NO), prostaglandin E2, myeloperoxidase (MPO), α-amylase, α-glucosidase, trypsin, a chymotrypsin kit, and an ELISA kit for interleukin-1β (IL-1β), interleukin-6 (IL-6), and tumor necrosis factor-α (TNF-α) were purchased from Nanjing Jiancheng Bioengineering Institute (Nanjing, China). 

### 2.2. Preparation of Foxtail Millet Protein Hydrolysate

The preparation of foxtail millet protein hydrolysate (FPH) was performed according to our previously published methods [19]. Briefly, ground millet flour was degreased with n-hexane. Alpha-amylase was added to remove the starch from defatted millet flour. After centrifugation, ultrapure water (1:50, *w*/*v*) was added to the pellet, and it was hydrolyzed with pepsin for 2 h (pH = 2.0, 37 °C). Next, trypsin was added and hydrolyzed for 2 h (pH = 7.5, 37 °C). The reactions were terminated by boiling and centrifuged at 9000× *g* for 15 min.

### 2.3. Degree of Hydrolysis, Protein Content, Molecular Weight Distribution, and Amino Acid Composition of FPH

Compositional analysis of the FPH was performed according to previously reported methods and our previous study. Briefly, the protein content of FPH was determined by the Folin reagent method [20]. The degree of hydrolysis (DH) of FPH was determined by the O-phthaldialdehyde (OPA) method [21,22]. The molecular weight (MW) of FPH was determined by HPLC, and the absorbance at 280 nm was detected using a TSK Gel G2000 SWXL chromatographic column (7.8 mm × 300 mm). Conalbumin (75,000 Da), ovalbumin (43,000 Da), cytochrome C (12,384 Da), aprotinin (6512 Da), vitamin B12 (1855 Da), and glutathione (307 Da) were used to establish the standard curve [23].

According to the method of the Chinese national standards (GB5009.124-2016), the composition of amino acids in FPH was determined using an amino acid analyzer [14].

### 2.4. Animal Grouping and Foxtail Millet Protein Hydrolysate Treatment

Male ICR mice were purchased from Weitong Lihua (Beijing, China) Laboratory Animal Technology Co., Ltd. Mice were raised in the Experimental Animal Center of Nankai University and maintained under standard laboratory conditions: 25 °C ± 2 °C and a 12 h light cycle. The experimental operation complied with the ethical requirements of animal experiments of Nankai University (permission number: SYKX 2019-0001).

#### 2.4.1. Protective Effect of Foxtail Millet Protein Hydrolysate on Ethanol-Induced Gastric Ulcers

The gastroprotective effect of FPH was evaluated using the ethanol-induced gastric ulcer model recommended by China’s Food and Drug Administration (CFDA, No. 107 in 2012). The mice were randomly divided into four groups, namely the normal group (NM), the model group (MD), the low-dose FPH group (LFPH), and the high-dose FPH group (HFPH), with 10 mice in each group and 3–4 mice per cage. Mice in the LFPH and HFPH groups were given 100 and 400 mg/kg·day of the FPH solution, respectively, while mice in the NM and MD groups were given an equal volume of saline for 14 days (Figure 1A). The dose was chosen on the basis of our previous study [19]. Twenty-four hours after the last gavage of FPH, mice were fasted with free access to water. The normal group was given 10 mL/kg saline by gavage, and the other groups were given 10 mL/kg ethanol. After one hour, mice were euthanized, and their stomachs, intestinal tissue, and serum were collected. The criteria for successful establishment of the model were gastric mucosal bleeding and linear blood spots.

#### 2.4.2. Protective Effect of Foxtail Millet Protein Hydrolysate on Pyloric Ligation-Induced Gastric Ulcers

To determine the gastric juice’s parameters, a mouse model of pyloric ligation-induced gastric ulcers was used [24]. Similarly, the mice were divided into four groups (NM, MD, HFPH, and LFPH) (*n* = 10). After 14 days of FPH gavage, the mice were fasted for 24 h and anesthetized by an intraperitoneal injection of a pentobarbital sodium solution (15 mg/mL, 2 mL). An abdominal incision was made and the pylorus was ligated according to previously reported methods [24]. Except for the NC group, all the other groups underwent pyloric ligation. The abdominal region was sutured, and the mice were euthanized 4 h later. Stomach and duodenum tissues, and gastric juice were collected.

#### 2.4.3. Gastroprotective Mechanism

One hundred mice were randomly divided into 10 groups (*n* = 10), and the mice were gavaged with 400 mg/kg FPH or the vehicle for 14 days (5 groups in each). Next, the abovementioned ethanol-induced gastric ulcer mouse model was established. Mice were pretreated with the blocking agents N-nitro-L-arginine methyl ester (L-NAME), N-ethylmaleimide (NEM), glibenclamide, and indomethacin before the ethanol gavage [25]. Each blocking agent corresponded to 2 groups of mice, one of which was the FPH gavage group and the other was the vehicle gavage group, for a total of 8 groups. The other two groups of mice were not pretreated with blocking agents and were set as the control group.

To investigate the role of endogenous prostaglandins (PGs) in the antiulcer effects of FPH, the mice were pretreated with indomethacin (10 mg/kg, dissolved in NaHCO_3_ and diluted in distilled water, i.p.) 30 min before ethanol-induced gastric ulceration. To investigate the role of endogenous NO in the gastric protection of FPH, the mice were pretreated with L-NAME (20 mg/kg, i.p.). To investigate the role of endogenous sulfhydryls (SHs) in the gastric protection of FPH, mice were pretreated with NEM (10 mg/kg, i.p.). To investigate the involvement of KATP channels in the gastroprotection of FPH, mice were pretreated with glibenclamide (5 mg/kg, i.p.) [25,26].

### 2.5. Evaluation of the Ulcerative Lesion Index and the Ulcer Inhibition Rate

The gastric tissue was incised along the greater curvature and rinsed with pre-cooled saline. Gastric mucosal ulcers were observed, and their length and width were measured. The values of the ulcerative lesion index (ULI) and the ULI inhibition rate were calculated according to Formulas (1) and (2), respectively [27].
ULI = 1 A + 2 B + 3 C(1)
where A represents the number of ulcers smaller than 1 mm, B represents the number of ulcers larger than 1 mm and smaller than 3 mm, and C represents the number of ulcers larger than 3 mm.
(2)$$\mathrm{Inhibition}\;\mathrm{rate}\;(\%)=[(ULI^{1}-ULI^{2})\times 100\%]/\;ULI^{1}$$
where *ULI*^1^ represents the ULI of the MD mice and *ULI*^2^ represents the ULI of the FPH-treated mice.

### 2.6. Histopathological Observations of the Gastric Tissues

After fixation with formalin, paraffin tissue sections (4 μm) were prepared and stained with hematoxylin–eosin (HE). Stained sections of the gastric mucosa were observed under a microscope and photographs were taken at 100× magnification.

### 2.7. Measurement of the Gastric Juice Parameters

After pyloric ligation, the gastric juice was collected and centrifuged at 8000× *g* for 10 min to obtain the gastric supernatant. The volume of gastric juice and the pH values were measured. Free acid and total acidity were determined by sodium hydroxide titration with methyl orange and phenolphthalein as the indicators [3]. Total acid output ([H+] mEq/mL/h) was calculated according to Formula (3).
Total acid output = (volume of gastric juice × total acidity)/4 h(3)

The activity of pepsin in the gastric juice was determined according to the kit’s instructions.

### 2.8. Determination of the Concentration of Gastric Mucus 

The concentration of gastric mucus was determined according to a previously published method [27]. Briefly, Alcian blue was used to stain the gastric tissue. The gastric tissue was rinsed with a sucrose solution to remove any excess Alcian blue. The complex mucus dye adhering to the stomach wall was extracted with a MgCl_2_ solution. Next, an equal volume of ether was added. After thorough mixing, the supernatant was removed by centrifugation at 3500× *g* for 20 min. The absorbance of the obtained emulsion was measured at 598 nm. The content of mucus was calculated with Alcian blue as a standard.

### 2.9. Biochemical Indexes in the Gastric Tissues

Gastric tissues were homogenized in cold phosphate buffered saline (pH 7.0, 1:10, *w*/*v*) [28]. A portion of the gastric tissue homogenate (10%, *w*/*v*) was centrifuged at 2500× *g* for 10 min to obtain the supernatant. The activity of MPO, CAT, GSH-PX, and SOD in the supernatant was determined according to the kit’s instructions. The levels of MDA, PGE2, and NO were determined following the manufacturer’s instructions. Another part of the gastric tissue homogenate was centrifuged at 5000× *g* for 15 min to determine the levels of inflammatory cytokines (IL-1β, IL-6, and TNF-α) using specific ELISA kits. The protein concentrations in the supernatant were measured by the BCA method.

### 2.10. Digestive Enzyme Activity in the Duodenum’s Contents

The duodenal contents of pylorus-ligated mice were collected and homogenized in cold phosphate buffered saline (pH 7.0, 1:10, *w*/*v*) [29]. After centrifugation for 15 min at 12,000× *g*, the supernatant was collected and used as the enzyme solution. The activities of α-amylase, α-glucosidase, trypsin, and chymotrypsin in the duodenal supernatant from the mice were determined following the kit’s instructions.

### 2.11. Western Blotting and qRT-PCR

Western blotting and qRT-PCR were performed according to our previously described methods [19]. The primers for RT-qPCR were purchased from Thermo Fisher (Shanghai, China). The gene-specific probes included MUC1 (Mm00449604_m1), MUC5AC (Mm01276718_m1), and MUC6 (Mm00725165_m1). The relative expression of the target genes was calculated by the 2-ΔΔCT method. For the Western blot analysis, the primary antibodies against eNOS (1:500), ET-1 (1:500), Akt (1:500), p-Akt (1:500), β-actin (1:1000), and the secondary antibody goat anti-rabbit IgG-HRP (1:5000) were used. The protein bands were formed by using enhanced chemiluminescence.

### 2.12. Statistical Analysis

All experimental results were expressed as the mean ± standard error, and statistical analysis was performed using Graph Pad Prism software. One-way ANOVA followed by Tukey’s multiple comparison tests and two-way ANOVA followed by Bonferroni’s post-hoc test were used to compare the differences among groups, with *p* < 0.05 indicating a significant difference.

## 3. Results

### 3.1. Ulcerative Lesion Index and Inhibition Rate

First, the gastroprotective effect of FPH was evaluated by using an ethanol-induced gastric ulcer model. The gastric mucosa of mice in the MD group was severely damaged, with obvious hemorrhage and linear blood spots (Figure 1B). FPH effectively improved gastric bleeding and ulcers. Compared with the MD group, the ULI of the mice in the HFPH group decreased significantly, and the ULI inhibition rate increased significantly (Figure 1C). The gastroprotective effect of 400 mg/kg FPH was better than that of 100 mg/kg FPH.

### 3.2. Inflammatory Response

MPO activity can reflect neutrophil infiltration and is a biomarker of mucosal injury and an inflammatory response. Ethanol induction significantly increased the activity of MPO and the expression levels of IL-1β, TNF-α, and IL-6 in the gastric tissue of mice (Figure 1C,D). The expression levels of IL-6, IL-1β, and TNF-α in the LFPH group were similar to those of the MD group, but the activity of MPO decreased significantly. The expression of IL-6, IL-1β, and TNF-α, and the activity of MPO decreased significantly in the HFPH group.

**Figure 1 antioxidants-11-02459-f001:**
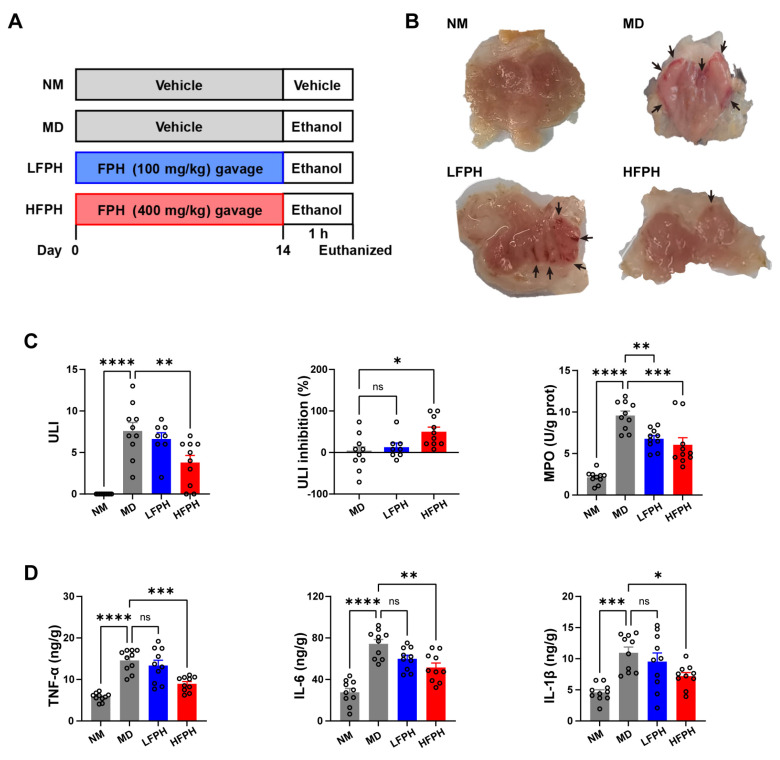
Gastroprotective effects of FPH on ethanol-induced gastric ulcers in mice. Protocol of the animal experiments (**A**). Macroscopic evaluation of gastric mucosal lesions (**B**). Ulcerative lesion index and gastric MPO activity (**C**). Levels of inflammatory cytokines in the gastric tissue (**D**). Data were expressed as the mean ± SEM. One-way ANOVA followed by Tukey’s multiple comparison tests (*n* = 10) was used to analyze the significance. * *p* < 0.05, ** *p* < 0.01, *** *p* < 0.001, and **** *p* < 0.0001 represent significant differences; ns, not significant.

### 3.3. Histopathological Analysis

Gastric tissue sections of the ethanol-induced gastric ulcers of mice were analyzed by HE staining. The mucosal layer, glandular layer, and submucosa of the NM group were intact (Figure 2A). However, in the MD group, the gastric mucosal epithelium was largely deficient, the structure of the glandular layer was severely disordered, the submucosa was marked with edema, and inflammatory spillover was observed. Compared with the MD mice, the degree of lesions in the LFPH mice was reduced, but the epithelial tissue was still obviously deficient, and the gland’s structure was partly disordered. The HFPH treatment had a significant inhibitory effect on gastric ulcer lesions. Only mild defects in the gastric mucosal epithelium and a small area of glandular structural disorder were observed.

### 3.4. Defensive Factors in Gastric Tissues

PGE2 inhibits the secretion of gastric acid and has immunosuppressive and anti-inflammatory effects. NO plays a crucial role not only in the regulation of gastrointestinal secretion and gastrointestinal motility but also in the inflammatory response as an immunoregulatory factor. Compared with the NM mice, the contents of PGE2 and NO in the gastric tissue of the MD mice were significantly reduced (Figure 2B). Treatment with FPH reversed this change in a dose-dependent manner.

### 3.5. Gastric Mucus Secretion

The gastric mucus contains mucin, bicarbonate, and other components, which can adhere to the surface of the mucous epithelium and isolate gastric acid. Compared with the NM mice, the gastric mucus content of the MD mice was significantly reduced (Figure 2B). The FPH treatment significantly reversed this change in a dose-dependent manner.

### 3.6. Antioxidant Activity

CAT is an endogenous reactive oxygen species-scavenging enzyme. SOD is an antioxidant metalloenzyme that catalyzes the radical transfer disproportionation reaction of superoxide anions. MDA is produced by lipid peroxidation and is one of the indicators commonly used to measure the degree of oxidative stress.

Ethanol treatment significantly decreased the activities of SOD, CAT, and GSH-PX, and increased the content of MDA in the gastric tissue of mice (Figure 2C). The activities of CAT, GSH-PX, and SOD in the LFPH mice were similar to those of the MD mice. The HFPH treatment significantly increased the activities of CAT, SOD, and GSH-PX. However, the FPH treatment did not change the MDA content.

### 3.7. Gastric Acid Secretion

Gastric acid is one of the main components of gastric juice, which is secreted by the parietal cells of the stomach. Pepsin is converted from pepsinogen under the action of gastric acid. An increase in the activity of pepsin indicates that excessive gastric acid secretion will damage the protective barrier of gastric mucosa, which may easily lead to chronic atrophic gastritis, gastric ulcers, and other diseases.

In the pylorus-ligated gastric ulcer model, the treatment with 400 mg/kg FPH significantly decreased the ULI value of the MD group and increased the ULI inhibition rate (Table 1). In addition, the pH value of the gastric juice decreased, and the gastric juice’s volume, total acidity, and total acid excretion significantly increased (Figure 3B). The pH value of the gastric juice in the HFPH group was similar to that of the MD group, but the total acidity and total acid excretion of the gastric juice were significantly decreased. Compared with the NM mice, the pepsin activity of the MD mice increased, and the HFPH treatment significantly reversed this change (Figure 3C).

### 3.8. Digestive Enzyme Activity

Pyloric ligation may lead to duodenal ulcers. Thus, we further determined the activity of digestive enzymes in the duodenal contents. Following pyloric ligation, the activity of trypsin was significantly increased in the duodenal contents of mice, but the activities of α-amylase, α-glucosidase, and chymotrypsin did not change significantly (Figure 3D). The FPH treatment dose-dependently decreased trypsin activity compared with the MD mice.

### 3.9. Effects of Pathway Blockers on the Gastric Protection of FPH

To explore the mechanism of the gastroprotective effect of FPH, we evaluated the effects of NO synthesis, SH compounds, PGE2, and KATP blocker pretreatments on the gastric prevention of FPH. FPH at 400 mg/kg had a protective effect on ethanol-induced gastric ulcers in the control mice that were not treated with the blocker (Figure 4B).

For SH compounds, the protective effect of FPH on gastric ulcers was ineffective after pretreatment with the blocking agent NEM. The value ULI was similar to that of the MD mice, and the level of ULI inhibition was significantly lower than that of the control group mice fed with FPH (Figure 4B). For NO synthesis, pretreatment with L-NAME also abolished the gastroprotective effect of FPH, manifested as similar ULI values to the MD mice, and significantly lowered the inhibition of ULI compared with the control mice. For the KATP channel, pretreatment with the blocker glyburide did not significantly affect the antiulcer effect of FPH. After the glyburide treatment, FPH still significantly reduced the ULI value of gastric ulcers in mice. Similarly, for PGE2 production, treatment with the blocker indomethacin did not significantly affect the gastroprotective effect of FPH.

Furthermore, FPH increased the mucin content in the gastric tissues of the control mice, but this effect was abolished by the NEM treatment (Figure 4D). Similarly, the increase in the NO content of the gastric tissue of FPH-treated mice was abolished by the L-NAME treatment (Figure 4E). In summary, these results suggested that NO and the mucous layer are key factors in the gastroprotective effect of FPH.

### 3.10. Effects of FPH on NO Synthesis and the ET-1/PI3K/AKT Pathway

We further investigated the upstream and downstream mechanisms related to NO synthesis. NO is produced by three isoenzymes, including endothelial nitric oxide synthase (eNOS), neuronal NOS (nNOS), and inducible NOS (iNOS). The first two isoforms are called constitutive NOS (cNOS), which produce the endogenous NO that maintains the integrity of the gastric mucosa and promotes wound healing [30]. Ethanol induction significantly decreased the total activity of NOS and the activity of cNOS in the gastric tissues of mice, but did not affect the activity of iNOS (Figure 5A). FPH significantly increased the total activity of NOS. Among them, the activity of cNOS decreased significantly, but the activity of iNOS did not change significantly. These results were further confirmed by Western blotting, in which FPH significantly increased the expression level of eNOS (Figure 5B).

Endogenous NO can inhibit the expression of endothelin-1 (ET-1) [31], which is an upstream effector of the PI3K/AKT signaling pathway and modulates the inflammatory response [32]. The Western blot results showed that ethanol induction significantly increased the expression of ET-1 and PI3K, and the phosphorylation level of Akt in the gastric tissues of mice (Figure 5C,D). Treatment with FPH decreased the expression levels of ET-1 and PI3K, and decreased the phosphorylation level of Akt. These results suggested that the anti-inflammatory effect of FPH is mediated by an increase in the activity of cNOS and inhibition of the ET-1/PI3K/Akt signaling pathway.

### 3.11. Effects of FPH on the mRNA Expression of Mucins

MUC1, MUC5AC, and MUC6 are the major mucins expressed in the gastric mucosa [33]. According to the results of qRT-PCR, the high-dose FPH treatment significantly increased the mRNA expression of MUC5AC and MUC6 in the gastric tissues of mice with ethanol-induced gastric ulcers (Figure 5E). However, neither ethanol nor FPH affected the expression of MUC1. These results suggested that FPH has a protective effect by increasing the expression of mucins.

### 3.12. Degrees of Hydrolysis, Molecular Weight Distribution, and Amino Acid Composition of FPH

The amino acid composition and molecular weight distribution play key roles in the biological activity of peptides [34]. Therefore, we further determined the degree of hydrolysis, the molecular weight distribution, and amino acid composition of FPH.

The protein content in FPH was 91.8% ± 5.95%, and the degree of hydrolysis was 32.6% ± 2.7%. Peptides with a molecular weight of <10 kDa accounted for 98.8% of the total content. Among them, peptides with molecular weights of 1–3 kDa had the highest content, accounting for 59.7%, and peptides with molecular weights of 3–5 kDa and 5–10 kDa accounted for 20.4% and 18.7%, respectively. Hydrophobic amino acids including Leu, Ala, Pro, Phe, Val, Ile, Tyr, and Met accounted for 50.2% of the total amino acids (Appendix A).

### 3.13. Prediction of Biopeptides in FPH

In total, 2620 peptide sequences were identified in FPH based on the results of UPLC-MS/MS in our previously published study [19], and these peptides were produced from 643 proteins.

In this study, key peptides in FPH were predicted using the BIOPEP protein database and the PeptideRanker score based on our previous LC-MS/MS data [19]. A PeptideRanker score of >0.2 and a relative abundance of >0.1% were set as the screening criteria. I total, 137 peptides were obtained, with lengths ranging from 6 to 23, of which 10 peptides had potential protective effects on the gastric mucosa (Table 2). The key sequences were PG and GP.

## 4. Discussion

Foxtail millet is widely cultivated in Asia and Africa, and is the most widely consumed coarse cereal in North China [35]. It has been traditionally thought to be beneficial for the stomach, but there is no experimental evidence to support its effectiveness. Protein is one of the main components of millet, and it has shown several biological activities, such as improving Type 2 diabetes and lipid metabolism in mice [36,37], and reducing hypertension in patients [16,38]. Millet-derived bioactive peptides have been demonstrated to have antioxidant activities [17]. Previously, we found that the preventive effect of millet protein on colitis was based on the regulation of intestinal mucin expression, which could be further explored for the management of gastrointestinal symptoms [19]. These previous studies suggested that FPH may have an antiulcer effect. Therefore, this study first evaluated the protective effects of FPH against gastric ulcers.

Ethanol induction and pyloric ligation are common models for studying gastric ulcers. Ethanol-induced gastric injury can cause mucosal ulcers, luminal hemorrhage, lipid peroxidative damage, and inflammatory activation [39]. The inflammatory response caused by gastric mucosal injury increases the expression of inflammatory cytokines including TNF-α, IL-1β, and IL-6 [40], and further triggers neutrophil infiltration and epithelial cell apoptosis, delaying the healing of gastric ulcers [41]. MPO is a biomarker that is indicative of neutrophil infiltration and inflammatory processes [3]. Stimulated neutrophils can release a large number of reactive oxygen species, leading to gastric mucosal damage. Antioxidative enzymes play a crucial role in the defense against oxidative damage to the gastric mucosa [28]. SOD is an antioxidant metalloenzyme that scavenges peroxyl free radicals. GPX protects gastric tissue from ROS damage by reducing lipid hydroperoxide. MDA reflects the tissue’s oxidative stress status and is associated with gastrointestinal ulceration and inflammatory damage [27]. Pyloric ligation exposes gastric cells to gastric acid by increasing vagal reflexes and stimulating the secretion of gastric acid, leading to gastric ulcers [42]. To evaluate the effects of FPH on oxidative stress, inflammation, gastric acid secretion, and digestive enzyme secretion during the progress of gastric ulcers, both ethanol- and pyloric ligation-induced mice models were used in this study. After 14 days of treatment, we found that FPH ameliorated gastric mucosal injury, decreased the activity of MPO, decreased the expression levels of cytokines, and increased the activity of antioxidant enzymes in the ethanol-induced gastric ulcer model. In the pyloric ligation model, FPH inhibited the secretion of gastric acid and decreased total acid output and the activity of pepsin in the gastric juice. The results suggested the gastroprotective effects of FPH. In addition, we used two doses (50 mg/kg and 200 mg/kg) in the study, consistent with our previous study [19]. The low dose of FPH only decreased the activity of MPO, but the high dose of FPH improved all the indicators related to oxidative stress and inflammation, indicating that FPH had a dose-dependent antiulcer effect. According to the body surface area model [43], the dose of 200 mg/kg FPH in mice was estimated to be the equivalent of 22 mg/kg in humans. This dose corresponds to a daily intake of approximately 1.5 g of millet protein and a daily intake of 8.3–13.6 g of foxtail millet. In this study, pepsin and trypsin were used to hydrolyze the millet protein to simulate natural digestion. Thus, the current results support the health effects of a millet-enriched diet.

Furthermore, the molecular mechanisms involving NO, mucus, KATP channels, and prostaglandins were analyzed by using blocking agents. Nitric oxide synthase (NOS) consists of eNOS, nNOS, and iNOS [30]. The first two enzymes constitute cNOS, which produces the endogenous NO that plays a crucial role in regulating gastrointestinal motility, mucosal function, blood flow, and inflammation. NO could downregulate the release of inflammatory mediators such as adhesion molecules from mast cells and reduce neutrophil adhesion [44]. It can increase the secretion of gastric mucus by increasing the levels of cGMP and accelerate wound healing through vasodilatory effects [29]. Studies have shown that NO has a gastroprotective effect and can accelerate the healing process of ulcer, while blocking NO synthesis by L-NAME aggravates the damage [45]. In our study, FPH treatment increased the expression of eNOS and the activity of cNOS, thereby increasing NO levels in the gastric tissue of mice. However, the gastroprotective effect of FPH decreased after L-NAME pretreatment. These results demonstrated that the increased level of endogenous NO is a key mechanism of the protective effect of FPH.

In addition, endogenous NO can inhibit the expression of ET-1 [31,46], which is the most effective vasoconstrictor. The increased secretion of ET-1 in the gastric mucosa can cause severe vasoconstriction under stress, which reduces the blood supply to the gastric tissue, leading to local hypoxia and acidosis [47]. ET-1 can also activate the PI3K/Akt signaling pathway, thereby activating the inflammatory response [32]. In this study, the FPH treatment decreased the expression of ET-1, PI3K, and the phosphorylation of Akt, indicating that the ET-1/PI3K/Akt signaling pathway played a role in the protective effect of FPH.

The mucus–bicarbonate barrier is the main defense system of the gastric mucosa. Mucus protects the viscosity of the gastric mucosa from irritating substances, invading pathogens, or mechanical damage. Due to its alkalinity, it prevents proteolysis by pepsin [48]. The secreted mucins MUCAC and MUC6 and the membrane-bound mucin MUC1 are the major mucins that make up the gastric mucus layer [33]. In addition, SHs bind to the mucus layer to form a stable protective barrier, preventing degradation by hydrochloric acid and proteolytic digestion by pepsin. The reduction of SHs makes the mucus soluble and more easily removed by harmful substances [49]. In this study, the staining results showed that the FPH treatment significantly increased the mucus content in the gastric mucosa, which supported our previous findings that FPH could increase the expression of mucins in the small intestine [19]. The antiulcer effects of FPH were attenuated after pretreatment of the mice with SHs inhibitors. Furthermore, FPH increased the mRNA expression of MUC5AC and MUC6 in the gastric mucosa. These results demonstrated that the gastroprotective effects of FPH are based on modulation of the mucus layer.

Endogenous prostaglandins such as PGE2 prevent the formation of gastric mucosal lesions by stimulating mucus secretion, increasing mucosal blood flow, and inhibiting gastric acid secretion [50]. PGE2 is continuously synthesized by COX in gastric epithelial cells, and its production can be blocked by indomethacin. Potassium-selective ion channels are pore-forming proteins that allow potassium ions (K^+^) to pass through the plasma membrane [51]. It is involved in smooth muscle cell relaxation and gastric mucosal defense, but glyburide inhibits this effect [52]. In the present study, the molecular mechanisms involving KATP channels and prostaglandins were investigated by using the blocking agents indomethacin and glibenclamide. Although the FPH treatment increased the expression of PGE2, pretreatment with indomethacin did not affect the antiulcer activity of FPH, indicating that COX and PGE2 are not key factors in the gastroprotective effect of FPH.

Similarly, FPH still exhibited antiulcer activity after pretreatment with the KATP channel inhibitor glibenclamide, suggesting that KATP channels are not the main driver of its gastroprotective effect.

The distribution of MW and the amino acid composition are considered to be critical factors affecting the bioactivity of peptides. Peptides with a MW< 3 kDa and with greater hydrophobicity have been suggested to be the bioactive peptides with the most potential [53]. In this study, 59.7% of the peptides in FPH had a molecular weight less than 3 kDa, and over 50% of the amino acids were hydrophobic, suggesting their potential biological activities. Moreover, PeptideRanker is a database for identifying peptides with possible bioactivity levels, including anti-inflammatory and antioxidative activities, which are correlated with the progression of gastric ulcers [54]. The content of biopeptides in FPH is positively related to the strength of the activity. BIOPEP is a database for predicting the activity of the parent peptide and its fragments. Based on the predictions by PeptideRanker and BIOPEP, and the relative peak area in LC-MS/MS, 10 peptides in FPH were suggested to exhibit potential protective effects in the gastric mucosa. The key sequences were PG and GP.

There are several limitations to this study. First, considering that FPH is a bioactive dietary ingredient, the gastroprotective effects of FPH were investigated by a comparison with the MD group rather than a positive control. However, a comparison with antiulcer agents, such as ranitidine, omeprazole, or cimetidine, would be helpful for verifying the gastroprotective effects of FPH or the bioactive peptides. Second, the antioxidant defense system in the gastric mucosa is composed of both enzymatic and nonenzymatic systems [54]. The former system includes SOD, CAT, GSH-PX, and glutathione reductase. The latter mainly includes the antioxidants GSH, thioredoxin, and melatonin. Here, we mainly focused on the activity of SOD, CAT, and GSH-PX. The determination of other endogenous enzymatic and nonenzymatic antioxidants would be helpful for better understanding the role of FPH.

## 5. Conclusions

In conclusion, we firstly demonstrated the protective effect of FPH against ulcerative colitis. FPH inhibited gastric mucosal lesions, decreased the expression of inflammatory cytokines, and improved the oxidative state of the gastric tissue. In addition, FPH increased the expression of mucins, the activity of cNOS, and NO levels, and inhibited the activation of the ET-1/PI3K/Akt pathway in the gastric tissues. Through pretreatment with blocking agents, we confirmed that the gastroprotective mechanism of FPH is the regulation of gastric mucosal mucus and NO synthesis. Our study provides strong evidence that millet protein, as a functional food, can improve gastric ulcers. Given that protein is a major component of millet, these findings support the health effects of a millet-enriched diet.

## Figures and Tables

**Figure 2 antioxidants-11-02459-f002:**
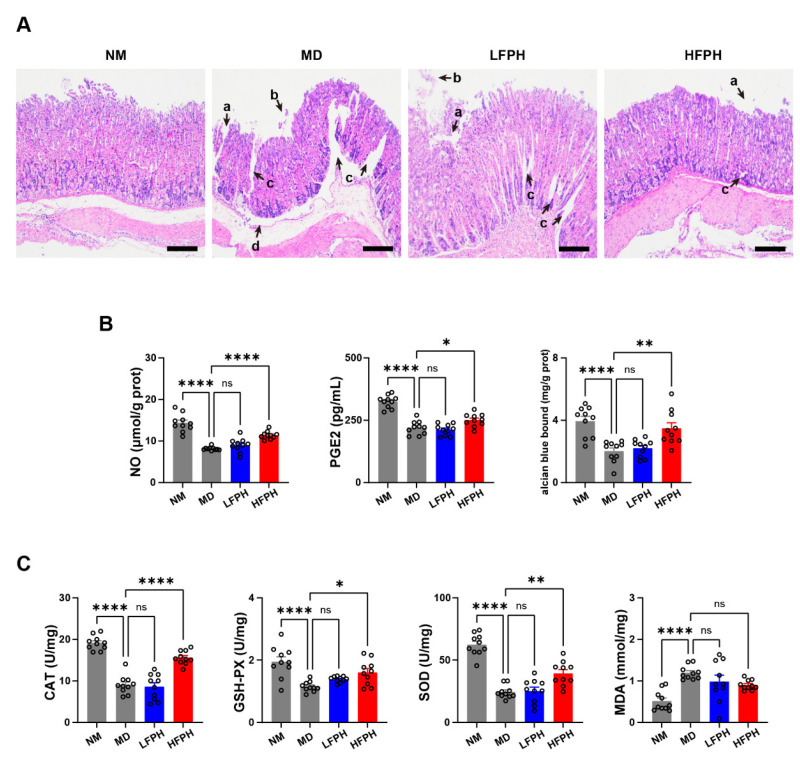
Effects of FPH on gastric histopathology, defensive factors, and antioxidant enzyme activity. HE staining of gastric mucosa sections. The arrows marked with “a” show mucosal epithelial defects, arrows marked with “b” show inflammatory spillover, arrows marked with “c” show glandular disorder, and arrows marked with “d” show submucosal edema (40× magnification, bar = 200 μm). (**A**). NO, PGE2, and mucin content in the gastric tissues (**B**). Antioxidant enzyme activities in gastric tissues (**C**). Antioxidant enzyme activities in the gastric tissues. Data were expressed as the mean ± SEM. One-way ANOVA followed by Tukey’s multiple comparison tests (*n* = 10) was used to analyze the significance. * *p* < 0.05, ** *p* < 0.01, and **** *p* < 0.0001 represent significant differences; ns, not significant.

**Figure 3 antioxidants-11-02459-f003:**
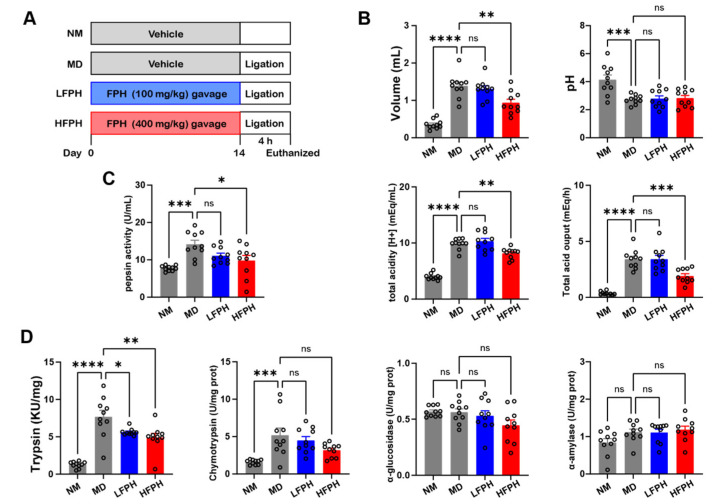
Effects of FPH on gastric acid and digestive enzyme secretion in mice with pyloric ligation-induced gastric ulcers. Protocol of the animal experiments (**A**). Gastric secretion parameters (**B**). Activity of pepsin in the gastric juice (**C**). Duodenal activity of digestive enzymes (**D**). Data are expressed as the mean ± SEM. one-way ANOVA followed by Tukey’s multiple comparison tests (*n* = 10) was used to analyze the significance. * *p* < 0.05, ** *p* < 0.01, *** *p* < 0.001, and **** *p* < 0.0001 represent significant differences; ns, not significant.

**Figure 4 antioxidants-11-02459-f004:**
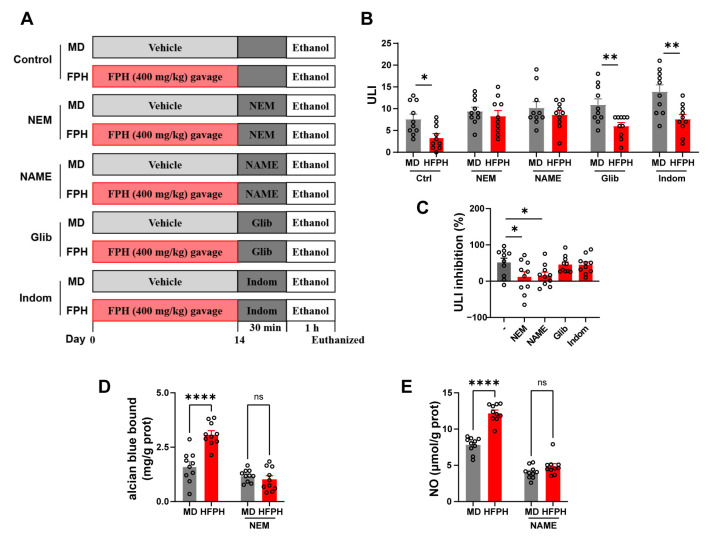
Effects of molecular pathway blockers on the gastroprotective effect of FPH. Protocol of the animal experiments (**A**). Ulcerative lesion index (ULI) (**B**). ULI inhibition (**C**). Mucin content in gastric tissues after the NEM treatment (**D**). NO content in gastric tissues after the NEM treatment (**E**). Data are expressed as the mean ± SEM. Two-way ANOVA followed by Bonferroni’s multiple comparison tests (*n* = 10) was used to test significance for (**B**,**D**,**E**). One-way ANOVA followed by Tukey’s multiple comparison tests (*n* = 10) was used to analyze the significance for (**C**). * *p* < 0.05, ** *p* < 0.01, and **** *p* < 0.0001 represent significant differences; ns, not significant.

**Figure 5 antioxidants-11-02459-f005:**
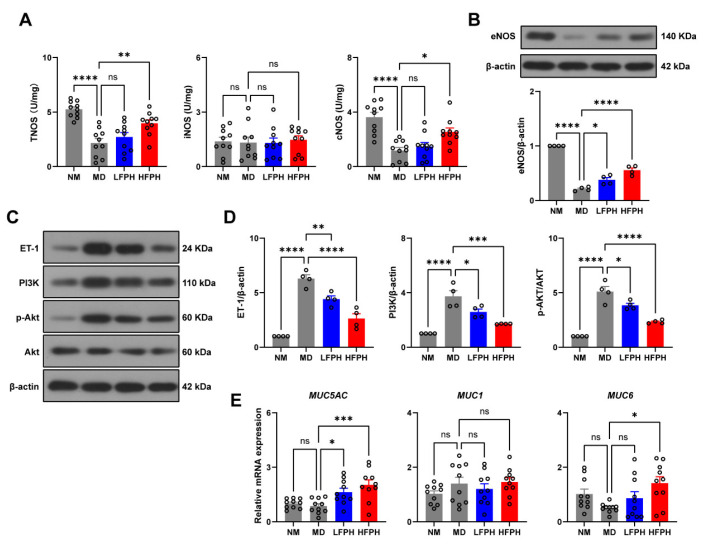
Molecular mechanisms of the gastroprotective effects of FPH on ethanol-induced gastric ulcers. Activity of NOS (**A**). Expression of eNOS (**B**). Expression of ET-1, PI3K, and Akt (**C**,**D**). mRNA expression of MUC5AC, MUC1, and MUC6 (**E**). Data are expressed as the mean ± SEM. One-way ANOVA followed by Tukey’s multiple comparison tests (*n* = 10 for (**A**,**E**); *n* = 4 for (**B**,**C**)) was used to analyze the significance. * *p* < 0.05, ** *p* < 0.01, *** *p* < 0.001, and **** *p* < 0.0001 represent significant differences; ns, not significant.

**Table 1 antioxidants-11-02459-t001:** Effects of FPH on pylorus ligation-induced ulcers in mice.

Group	Dose (mg/kg)	ULI	ULI Inhibition (%)
NM	-	0 ****	-
MD	-	8.19 ± 1.04	0 ± 12.7
LFPH	100	7.20 ± 0.87	12.1 ± 10.6
HFPH	400	4.96 ± 0.74 *	39.4 ± 9.08 *

Data are expressed as the mean ± SEM. One-way ANOVA followed by Tukey’s multiple comparison tests (*n* = 10) was used to analyze the significance. * *p* < 0.05 and **** *p* < 0.0001 represent significant differences compared with the MD group.

**Table 2 antioxidants-11-02459-t002:** Potential bioactive peptides in FPH.

Parent Protein ^a^	Peptide Sequence	Relative Peak Area (%)	Retention Time (min)	Experimental *m*/*z*	Error (ppm)	PeptideRanker Score ^b^	Potential Bioactive Peptides ^c^
K3XLT8	IDFAPGGQNPPHTHPR	5.67	17.8	870.9341	2.7	0.55	PG
K4A875	HASEGGHGPHWPLPPFGES	1.76	34.3	998.4583	−0.2	0.49	GP
K4A875	HASEGGHGPHWPLPPFGESHGPY	1.46	33.1	1225.5571	0.4	0.47	GP
K3XLT8	IDFAPGGQNPPHTHP	14.3	22.8	528.9232	−0.3	0.46	PG
K4A875	HASEGGHGPHWPLPPFGESHGP	1.27	29.3	1144.0265	1.3	0.41	GP
K3XW03	SLGVAGSQPGIEGEEIAPL	0.14	54.9	912.4725	−0.3	0.35	PG
K3XEI1	GHVFEEMQRPGTPL	0.10	29.2	799.3934	0.3	0.33	PG
K3Z668	VETGIIKPGM	0.14	24.5	522.7918	0.6	0.32	PG
K3Z668	NHPGQIGNGYAPV	1.31	22.5	662.3253	−0.6	0.31	PG
K3Z668	VIIMNHPGQIGNGYAPV	0.15	41.3	890.4644	0.3	0.24	PG

^a^ From Uniprot (https://www.uniprot.org/ (accessed on 16 January 2022)). ^b^ From PeptideRanker (http://distilldeep.ucd.ie/PeptideRanker/ (accessed on 23 January 2022)). ^c^ From BIOPEP (https://biochemia.uwm.edu.pl/biopep-uwm/ (accessed on 27 January 2022)).

## Data Availability

Data are contained within the article.

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
