# Peer review of "Protective Effect of Foxtail Millet Protein Hydrolysate on Ethanol and Pyloric Ligation-Induced Gastric Ulcers in Mice"

_antioxidants, 2022, doi:10.3390/antiox11122459_

Round 1

Reviewer 1 Report

The report describes the protective effect of foxtail millet protein hydrolyzate on ethanol and pyloric ligation-induced gastric ulcer in mice.

 According to the authors, the anti-ulcer activity of millet has not been studied; introduction lines 64-65 “These results suggest that millet protein may have protective effects on gastric ulcers, but its activity has not been studied.” However, the anti-ulcer activity of this plant was previously reported by H.-C. Lin et al. (Journal of Traditional and Complementary Medicine 10, 2020, 336-344). However, the present work reports new information on the gastroprotective activity of foxtail millet protein.

 The work in general is good. However, ¿why the authors will not use positive control in the gastro protective models? Like: ranitidine, omeprazole or cimetidine. The positive control is necessary to allow validation of the results obtained.

 I have only found minor errors, please Check:

Lines 253-254: “Compared with the NM mice, the contents of PGE2 and NO in the gastric tissue of the MD mice were significantly increased (Figure 2B).” correct for: were significantly reduced.

Line 300: “FPH treatment dose-dependently decreased trypsin activity”. The full sentence should be: FPH treatment dose-dependently decreased trypsin activity compared to MD.

There is a large number of studies on millet in the literature. Many of them on antioxidant activity that could have been cited.

Reviewer 2 Report

In the present study, entitled “Protective effect of foxtail millet protein hydrolyzate on ethanol and pyloric ligation-induced gastric ulcer in mice” by Zhang et al., the authors used two gastric ulcer models, ethanol- and pyloric ligation-induced, to investigate the protective effect of foxtail millet protein hydrolyzate (FPH) on stomach in mice. They also applied several inhibitors or blockers to elucidate the possible mechanism of protective effect of FPH. They found that FPH obviously attenuated the gastric ulcer caused by ethanol or pyloric ligation due to the regulation of gastric mucosal mucus and NO synthesis and the inhibition of ET-1/PI3K/Akt pathway. The authors further identified the peptides in FPH and found that the key sequences were GP and PF. Basically, this study did a lot of work and found some interesting results. However, there are still several issues with this study, as follows:

1.      The authors just determined the activities of GSH-PX, CAT, and SOD in the antioxidant system of gastric mucosa, while the content of GSH and the activity of GSH-RD are closely related to these enzymes. Why didn't the authors analyze the content of GSH and the activity of GSH-RD in the present study?

2.      For the ULI inhibition ratio of FPH showed in Figure 1C, the NM group should not with an ULI inhibition ratio since there is not ULI in NM group.

3.      In the section of Discussion, the authors spend a lot of content to describe the information in the literature, but rarely discuss or compare the effects of their samples with this information. I think the authors should rewrite or modify it here.

4.      Also in the section of Discussion, the authors did not mention their predicted peptide sequences at all. The authors are requested to provide additional narrative here.

5.      It's a weird format to break the abstract into two paragraphs in the article, please fix it.

6.      Line 38, the “Helicobacter pylori” should be italicized.

7.      Line 172, the 2 of MgCl2 should be subscripted.

8.      Line 204, the p of p<0.05 should be italicized.

9.      Why is there a millet-enriched diet.6. Patents at the end of the conclusion?
